# Low-Concentration T-2 Toxin Attenuates Pseudorabies Virus Replication in Porcine Kidney 15 Cells

**DOI:** 10.3390/toxins14020121

**Published:** 2022-02-06

**Authors:** Kuankuan Xiong, Lei Tan, Siliang Yi, Yingxin Wu, Yi Hu, Aibing Wang, Lingchen Yang

**Affiliations:** Laboratory of Animal Disease Prevention & Control and Animal Model, Hunan Provincial Key Laboratory of Protein Engineering in Animal Vaccines, College of Veterinary Medicine, Hunan Agricultural University (HUNAU), Changsha 410125, China; kuankuanxiong@stu.hunau.edu.cn (K.X.); leitan@stu.hunau.edu.cn (L.T.); Y.S.L@stu.hunau.edu.cn (S.Y.); yingxinwu@stu.hunau.edu.cn (Y.W.); yihu@stu.hunau.edu.cn (Y.H.); bingaiwang@hunau.edu.cn (A.W.)

**Keywords:** T-2 toxin, PRV, PK15 cell, oxidative stress, apoptosis

## Abstract

Pseudorabies, caused by pseudorabies virus (PRV), is the main highly infectious disease that severely affects the pig industry globally. T-2 toxin (T2), a significant mycotoxin, is widely spread in food and feeds and shows high toxicity to mammals. The potential mechanism of the interaction between viruses and toxins is of great research value because revealing this mechanism may provide new ideas for their joint prevention and control. In this study, we investigated the effect of T2 on PRV replication and the mechanism of action. The results showed that at a low dose (10 nM), T2 had no significant effect on porcine kidney 15 (PK15) cell viability. However, this T2 concentration alleviated PRV-induced cell injury and increased cell survival time. Additionally, the number of PK15 cells infected with PRV significantly reduced by T2 treatment. Similarly, T2 significantly decreased the copy number of PRV. Investigation of the mechanism revealed that 10 nM T2 significantly inhibits PRV replication and leads to downregulation of oxidative stress- and apoptosis-related genes. These results suggest that oxidative stress and apoptosis are involved in the inhibition of PRV replication in PK15 cells by low-concentration T2. Taken together, we demonstrated the protective effects of T2 against PRV infection. A low T2 concentration inhibited the replication of PRV in PK15 cells, and this process was accompanied by downregulation of the oxidative stress and apoptosis signaling pathways. Our findings partly explain the interaction mechanism between T2 and PRV, relating to oxidative stress and apoptosis, though further research is required.

## 1. Introduction

Pseudorabies virus (PRV), also called Suid herpesvirus 1, is an enveloped, double-stranded linear DNA virus belonging to the subfamily *Alphaherpesvirinae*, within the *Herpesviridae* family [1]. PRV can infect a wide variety of animals, including pigs, ruminants (sheep/goat and cattle/cows), carnivores (minks and foxes), and rodents, though pigs, including wild boars, are the natural host for this pathogen [2]. Recently, a live PRV strain was successfully isolated from a patient with acute encephalitis, and its potential cross-species transmission ability has drawn wide attention [3]. PRV remains widely prevalent in numerous counties and regions, posing a huge threat to the pig industry. The clinical symptoms of pseudorabies are diarrhea, vomiting, nervous system disorders with high mortality among suckling piglets, dyspnea and slow growth of fattening pigs, and reproductive disorders in breeding pigs [4]. PRV infects porcine kidney 15 (PK15) cells and causes cytopathy and cell damage. The virus infection also causes oxidative stress, affects the activity of antioxidant enzymes, and induces the generation of reactive oxygen species (ROS), further damaging cell components and even leading to cell death [5]. PRV-induced oxidative stress activates Bcl-2-family-and caspase-family-related proteins, further leading to apoptosis and DNA damage [6].

T-2 toxin (T2) is one of the most prevalent natural pollutants in field crops, including wheat, corn, barley, and livestock feeds [7,8]. Wang et al. detected T2in grain from Sichuan, China, in 2012 and found that the detection rate of T2 was 11.64% and the average dose was 0.565 µg·kg^−1^. The maximum doses of T2were 3.332 µg·kg^−1^. The incidence of T2in grains and agricultural products is lower than that of related mycotoxins, such as deoxynivalenol (DON) [9]. A recent study showed that 5289 of 27,850 feed specimens collected worldwide were polluted with T2. The contamination rates of this toxin in feedstuffs or field crops in various regions have been reported in previous studies [10]. General signs of T2 toxicity in animals include weight loss, feed refusal, vomiting, rash, bleeding, diarrhea, and even death [11]. It has been universally confirmed that persistent exposure to T2 causes anabrosis and necrosis of oral and intestinal mucosa in pigs, leading to vomiting, diarrhea, reduction in feed intake, and other effects, such as immunotoxicity [12]. Additionally, T2 induced oxidative stress and DNA damage with a time-dependent significant increase in p53 expression and activation of the apoptotic process via caspase-3 in human cervical cancer cells [13]. Furthermore, both T2 and HT-2 toxin induced apoptosis in chondrocytes by increasing oxidative stress, which caused the activation of Bax, caspase-3, and caspase-9 [14].

The interactions between viruses and toxins and the underlying mechanisms are of great interest. Aflatoxin B1 (AFB1) is the most carcinogenic mycotoxin. It was found in 1996 that human hepatitis B virus (HBV) and AFB1 play a synergistic role in the development of tumors [15]. AFB1 is also a cofactor in the human herpesvirus-type-4-mediated carcinogenesis effect. Similarly, T2 aggravated the infection effects of avian infectious nasal bronchitis virus (IBV [16]), enhanced the anticoagulant effect of avian Newcastle disease virus (NDV [17]), and weakened the immune response of mice to reovirus (REOV), thus aggravating viral bronchitis [18]. In contrast, deoxynivalenol (DON) significantly reduced the replication of porcine reproductive and respiratory syndrome virus (PRRSV) in vitro and in vivo [19], reduced the immune response of pigs to PRRSV, and affected the infection process of PRRSV [20]. Additionally, pig feed contaminated with DON impaired the immune response induced by an attenuated PRRSV vaccine [21]. Recent studies on porcine circovirus type 2 (PCV2) and ochratoxin A (OTA) have found that oxidative stress and autophagy induced by OTA promote PCV2 replication in vivo and in vitro [22]. Overexpression of selenoprotein S (SELs) inhibited oxidative stress and p38 phosphorylation in PK15 cells and blocked the OTA pro-replication effect on PCV2 [23]. Furthermore, selenomethionine (SeMet) inhibited autophagy by activating the Akt/mammalian target of rapamycin (mTOR) signaling pathway, thereby reducing the replication of PCV2 [24]. It has also been shown that PCV2 enhances OTA-induced nephrotoxicity through p38-mediated autophagy during infection. Chicken inclusion body hepatitis virus (IBHV) and OTA also play a synergistic role in aggravating anemia in chickens [25].

With the rapid development of the pig industry in China, PRV infection and T2 pollution of feeds are more common in this industry; however, there are few reports on the effects of T2 and PRV in vivo and in vitro. In this study, we investigated the effects of T2 on PRV infection and replication and examined whether the effects are related to oxidative stress and apoptosis. In this study, we also investigated the interaction between T2 and PRV in vitro and the signaling pathways affected by this interaction to provide new insight into the potential relationship between toxins and viruses and provide a theoretical basis for the prevention and control of animal diseases.

## 2. Results

### 2.1. T2 Cytotoxicity

We found that the cell activity of PK15 cells treated with various T2 concentrations decreased in a dose-dependent manner at 24 and 48 h, and cell activity was the lowest when 100 nM T2 was used (*p* < 0.001) (Figure 1). The IC_50_ of T-2 toxin to PK15 cells was 78.19 nM. Because 10 nM T2 had no significant effect on PK15 cell viability at 24 and 48 h (*p* > 0.05), this concentration was selected for the following experiments examining the interaction between T2 and PRV. We performed MTT assay by measuring the absorbance of MTT at 490 nm and compared the activity in PK15 cells to a DMEM control.

### 2.2. T2 Inhibits the Cytotoxicity and Cell Death Caused by PRV Infection

We observed no significant difference in cellular morphology between the T2 and NC groups (Figure 2). However, the PRV-infected (MOI = 0.01, same as below) cells were rounded and shedding and showed obvious shrinkage. Interestingly, a significant decrease in the cytopathy of PK15 cells infected with PRV was observed 24 h after T2 exposure.

#### 2.2.1. The Number of Cells Infected by PRV

The number of PRV-infected cells was determined by immunofluorescence assay. PRV gB protein was detected by immunofluorescence assay at 20 h after PRV infection, with MOI = 0.01 in all experimental groups. Red fluorescence indicated cells that were infected with PRV. No fluorescence was observed in the NC group, whereas in the PRV group, about 80% of the cells were infected. When T2 was used, 5 and 10 nM T2 significantly decreased the number of PRV-infected cells (*p* < 0.05) (Figure 3).

Next, we determined the PRV (MOI = 0.01) copies in T2-treated and untreated PK15 cells. Total DNA of the PRV-infected cells was extracted, and the virus copy number was determined by quantitative PCR (qPCR). Compared with cells in the PRV group, the number of PRV copies in the PRV+T2 group significantly decreased from 10 to 24 h (*p* < 0.05) (Figure 4). However, there was no significant difference between these two groups after 36 h (*p* > 0.05).

#### 2.2.2. The Mode of Action Assay

When we investigated the T2 mode of action, we performed four assays (virus inactivation assay, virus absorption assay, pre-treatment assay, and virus replication assay) to investigate the effects of T2 on PRV infection in PK15 cells, and finally the viral copies in each group were determined by qPCR. The results showed that there was no significant difference in the PRV copy number between the T2-treated and untreated cells in the PRV inactivation assay, pre-treatment assay, and virus entry assay (*p* > 0.05). However, in the virus replication assay, the PRV copy number in the T2-treated PK15 cells was significantly lower than that in the untreated cells (*p* < 0.001). The above results suggested that T2 interferes with PRV infection in vitro mainly via inhibiting viral replication, but not viral inactivation, absorption, and entry (Figure 5).

### 2.3. Oxidative Stress and Apoptosis Are Involved in the Inhibition of PRV Replication in PK15 Cells by Low-Concentration T2

To evaluate oxidative stress in PRV-infected cells treated with T2, we measured the mRNA expression of *Nrf2*, *Keap1*, *Gpx-1*, and *Nqo1* in the four indicated groups (Figure 6).

Nrf2 and Keap1 are key factors of related antioxidant signaling pathways, and Gpx-1 and Nqo1 are key genes of related antioxidant enzymes. Compared with the NC group, the mRNA levels of *Nrf2*, *Keap1*, and *Nqo1* in the T2-treated cells were significantly downregulated (*p* < 0.05) (Figure 6A,B,D). However, the four genes mentioned above were significantly upregulated in the PRV group compared with the NC group (*p* < 0.001) (Figure 6A–D). Interestingly, compared with the PRV group, the mRNA levels of these genes mentioned above were markedly downregulated in the PRV+T2 group (*p* < 0.05) (Figure 6A–D).

Bcl-2 and Bax belong to a family that regulates apoptosis activators by controlling the permeability of the mitochondrial membrane. The caspase pathway is one of the main pathways of apoptosis. When we evaluated the effect on apoptosis-related genes, we found that compared with the NC group, *caspase-3*, *caspase-8*, and *Bax* mRNA expression was significantly upregulated in the PRV group (*p* < 0.01) (Figure 7A–C). However, the mRNA expression of *Bcl-2* was significantly downregulated in the PRV group (*p* < 0.01) (Figure 7D). Additionally, compared with the PRV group, the mRNA expression levels of *caspase-3*, *caspase-8*, and *Bax* in the PRV+T2 group were significantly lower (*p* < 0.01) (Figure 7A–C), whereas the mRNA expression of *Bcl-2* was significantly upregulated (*p* < 0.01) (Figure 7D).

To verify that oxidative stress and apoptosis are involved in the inhibition of PRV replication in PK15 cells by low-concentration T2, we examined the protein expression of cleaved-caspase-3, cleaved-caspase-8, Bax, Bcl-2, Nrf2, and GPx-1 by Western blotting. As shown in Figure 8, compared with the NC group, the proteins expression levels of cleaved-caspase-3, cleaved-caspase-8, and Nrf2 and the ratio of Bax/Bcl-2 in the PRV group were significantly upregulated (*p* < 0.01), and the proteins expression levels of Gpx-1 in the PRV group were significantly upregulated (*p* < 0.05). Additionally, compared to the T2 treatment group, the proteins expression levels of cleaved-caspase-8 and Nrf2 and the ratio of Bax/Bcl-2 in PRV-infected PK15 cells were significantly downregulated (*p* < 0.01); furthermore, the protein expression levels of cleaved-caspase-3 and Gpx-1 were markedly downregulated (*p* < 0.05).

## 3. Discussion

Mycotoxins are often detected in grains from different sources that are used to feed animals. The mycotoxin T2 is widely found in feed, which poses a great threat to the healthy growth of pigs. The occurrence and prevalence of porcine viral diseases are related to many factors, including feed toxin residues [26]; T2 can cause anorexia, weight loss, vomiting, rash, bleeding, diarrhea, and even death. It can also lead to the inhibition of protein, DNA, and RNA synthesis in animal cells, resulting in immunosuppression, oxidative stress, and apoptosis, which lead to cell damage [12]. The toxic dose of T2 in pigs is different in vivo and in vitro, just as the toxic concentration of T2 in pigs and PK15 cells is different [27,28]. Even the toxic concentration of T2 for PK15 cells and primary porcine alveolar macrophages (PAMs) of the same pig origin is different [29]. The physiological concentration of T2 is different in different animals and cells.

PRV is considered one of the most important pathogens in pigs, and pigs are the only natural host of PRV infection [30]. PRV infects PK15 cells and causes cytopathy and cell damage. This virus also causes oxidative stress, affects the activity of antioxidant enzymes, and induces ROS production, which further damage cell components and even lead to cell death. Although both T2 and PRV cause oxidative stress in cells, there is no report on the role of T2 in the process of PRV infection and the related mechanism. Therefore, we explored the effect of T2 on PRV infection and the replication in host cells and the potential mechanism. We did this to explain the different characteristics and the severity of PRV occurrence on pig farms, thereby providing new ideas for the prevention and control of PRV, to promote the healthy, stable, and rapid development of the pig industry.

In our study, the activity of PK15 cells decreased upon T2 treatment in a dosage-dependent manner. However, we found that 10 nM T2 had no significant effect on the activity of PK15 cells, which is consistent with other studies on primary pig cells [31]. Therefore, we used 10 nM T2 to examine the effect of T2 on PRV infection. In the PK15 cell model used in this study, PRV caused obvious cytopathy at 12 h, and most of the cells died of PRV infection within 36 h. When we examined the PK15 cell morphology, PRV caused obvious cytopathy and led to cell death. Interestingly, PRV-infected PK15 cells exposed to a safe concentration (10 nM) of T2 exhibited significantly reduced cytopathy and prolonged survival time. Our immunofluorescence data also showed that the number of PRV-infected cells that were exposed to a safe concentration (10 nM) of T2 significantly decreased. Therefore, we speculated that the low T2 concentration (10 nM) inhibits PRV replication in PK15 cells. In the following qPCR analysis of the PRV copy number, we confirmed this conjecture. Examining the PRV copy number in PK15 cells revealed that 10 nM T2 inhibits the proliferation efficiency of PRV in PK15 cells, which was more obvious in the early stage postinfection (up to 24 h). After 24 h, there was no significant difference in the PRV copy number between T2-treated and untreated cells, possibly because of the obvious pathological changes in PK15 cells without T2 treatment and the weakened replication ability of PRV. Therefore, the T2 inhibitory effect on PRV proliferation mainly occurred during the early incubation period. After 36 h of incubation, the PK15 cells of the group without T2 treatment were significantly affected by PRV infection and a large number of the cells died. In general, T2 inhibited the replication efficiency of PRV, which explains the decreased cytopathy and cell death in PRV-infected PK15 cells. When we evaluated the mode of action of T2 on PRV in PK15 cells, we found that among the four modes of action tested, T2 treatment significantly decreased the PRV copy number in PK15 cells only in the virus replication assay. This illustrated that T2 causes intracellular inhibition of PRV replication. To the best of our knowledge, this is the first study to report that T2 inhibits PRV replication at the cellular level, contrary to the previously reported immunosuppressive effect of T2 on the avian infectious disease induced by IBV infection [16]. However, our finding is similar to the result that the vomiting toxin DON significantly reduces the replication of PRRSV in vitro and in vivo [19].

Because both T2 and PRV induce oxidative stress damage accompanied by apoptosis, we investigated whether the T2 inhibition of PRV replication is related to oxidative stress and apoptosis by determining the expression level of cytokines related to oxidative stress and apoptosis. The results showed that 10 nM T2 significantly downregulates the expression of oxidative-stress- and apoptosis-related genes, suggesting that oxidative stress and apoptosis are involved in the inhibition of PRV replication in PK15 cells by low-concentration T2 and may be related to the Nrf2 signaling pathway. These results are similar to those of a study on the cytotoxicity of T2 in HepG2 cells, where the exposure of HepG2 cells to nontoxic concentrations of T2 protected the cells against subsequent cellular oxidative conditions induced by even higher concentrations of the mycotoxin [32].

Although high concentrations of T2 cause apoptosis and oxidative stress [30], we found that a low T2 concentration inhibits PRV replication and downregulates oxidative stress and apoptosis-related genes. Our preliminarily findings of the involvement of oxidative stress and apoptosis in the protective mechanism of T2 provide new ideas for developing new PRV treatments and prevention programs for pig farms. Our study also provides new findings on the T2 effects and mechanism. Finally, this study provides a new scientific reference and theoretical basis for the study of the interaction between viruses and toxins and the potential mechanism.

## 4. Conclusions

In summary, we revealed that although 10 nM T2 had no effect on cell activity, it reduced cell damage and cell death induced by PRV infection in PK15 cells. Our results showed that a low T2 concentration inhibits PRV replication in PK15 cells and that oxidative stress and apoptosis are involved in this inhibition.

## 5. Materials and Methods

### 5.1. Cells and Viruses

PK15 cells were cultured in Dulbecco’s modified Eagle Medium (DMEM) supplemented with fetal bovine serum (FBS), penicillin, and streptomycin at 37 °C with 5% CO_2_. The PRV strain was isolated from tonsil tissues of dead piglets previously and stored in our lab, which was genetically close to the PRV classical strains (such as Ea and Fa) prevalent in China. Further sequence analysis showed that the isolates were genetically close to the PRV traditional and variant strains. The PRV strains were propagated in PK15 cells and stored at −80 °C.

### 5.2. Reagents and Antibodies

T2 was obtained from Sigma-Aldrich (St. Louis, MO, USA) and dissolved in DMEM at 10 µM. Primary antibodies (rabbit anti-β-actin pAb (CST, Danvers, MA, USA) and mouse anti-IκBα mAb (Proteintech, Rosemont, IL, USA)) and secondary antibodies applied for Western blotting (goat anti-mouse IgG (H+L) (Promega, Madison, WI, USA) and goat anti-rabbit IgG (H+L) (Promega, Madison, WI, USA)) were purchased from commercial companies. Cyanine 5(Cy5)-conjugated goat anti-mouse IgG used in confocal microscopy was obtained from Gibco BRL life Technologies (New York City, NY, USA). Mouse monoclonal anti-cleaved-caspase-3 antibody and anti-cleaved-caspase-8 were purchased from Servicebio (Hubei, Wuhan, China). Rabbit polyclonal anti-Nrf2 antibody, anti-Gpx-1, mouse monoclonal anti-Bax antibody, and anti-Bcl-2 were purchased from Bioss (Beijing, China). In addition, the primary antibody mouse anti-PRV-gB mAb was a gift obtained from Dr. Ping Jiang (College of Veterinary Medicine, Nanjing Agricultural University, Nanjing, China).

### 5.3. Cell Cytotoxicity Assay

PK15 cells were cultured in 96-well plates. At approximately 90% confluence, the cells were treated with different concentrations of T2 or DMEM for 24 or 48 h. Subsequently, the cells were incubated with 10 µL of MTT solution for 4 h. Finally, the supernatants were removed, and 150 µL of DMSO was added and the cells were again incubated for 10 min. The cell activity was determined by analyzing the mean absorbance values (OD490 nm) of six wells using a spectrophotometer. Cell activity = (OD490 (treatment)/OD490 (control)) × 100%. The IC_50_ value of T-2 toxin was calculated using GraphPad Prism 8 software. All the above procedures were performed under aseptic conditions.

### 5.4. Morphological Investigation

PK15 cells were divided into four groups: blank control group (NC group), T2 treatment group (T2 group), PRV infection group (PRV group), and T2 plus PRV treatment group (T2 + PRV group). The cells were cultured in 6-well plates. The NC group cells were cultured in DMEM. The T2 group cells were cultured with 10 nM T2. The PRV group cells were infected with PRV (multiplicity of infection (MOI) = 0.01). The T2+PRV group cells were infected with PRV (MOI = 0.01) for 2 h, and then the PRV inoculum was removed and the monolayer of PK15 cells was washed three times with PBS. Next, these cells were incubated with 10 nM T2. The cells were observed under a microscope at 16, 24, and 36 h.

### 5.5. Immunofluorescence Staining of Glycoprotein B (gB)

PK15 cells were cultured in 6-well plates. When the cell density reached 70–80%, the cells were infected with PRV (MOI = 0.01). After 2 h, the PRV inoculum was removed and the cell monolayer was washed three times with PBS. Next, the cells were treated with different T2 concentrations or DMEM for 20 h. The cells were examined for gB expression using immunofluorescence following the manufacturer’s instructions. Briefly, the cells were incubated with antibodies for the PRV gB protein overnight at 4 °C, followed by three washes with PBS and incubation with goat anti-rabbit IgG secondary antibodies. The cells were counterstained with CY5 for 10 min and washed with PBS. Fluorescence was imaged under a Zeiss LSM 510 laser confocal microscope (Precise, Beijing, China). Cells positive for PRV gB were counted in six fields. The calculation of the relative proportion of infected cells was based on the total amount of cells in each field.

### 5.6. Determination of PRV DNA Copies by Real-Time PCR (qPCR)

To explore the effect of T2 on PRV infection, PRV (MOI = 0.01) was used to inoculate cells in a 6-well plate with or without 10 nM T2. Total DNA of the PRV-infected cells was extracted, and the virus copy number was determined by quantitative PCR (qPCR). The standard curve was as follows: log(virus copy number) = −3.2428 Ct + 12.084 (*R*^2^ = 0.995). The viral DNA in the PRV-infected cells and in the supernatants was extracted using commercial DNA extraction kits following the manufacturer’s instructions. The number of PRV DNA copies in the PK15 cells was determined by an RT-PCR kit (Accurate Biology, Changsha, China), as described previously. In brief, DNA extraction was performed using a DNA Mini kit (Accurate Biology, Changsha, China) and the purified DNA was used as a template for SYBR Green RT-PCR amplification. A 125 bp fragment of the PRV *gB* gene was amplified using forward and reverse primers (the prime sequences targeting to different genes are shown in Table 1). SYBR Green RT-PCR was performed using the ABI Prism Step One Plus detection system (Applied Biosystems, Shanghai, China). A recombinant PRV19 plasmid vector containing a PRV genome insertion was used as the standard.

### 5.7. The Mode of Action Assays

PK15 cells were divided into the following four groups to analyze the mode of action [33]:

PRV inactivation assay: The infectious viruses were pre-incubated in DMEM with or without T2 for 2 h at room temperature. PK15 cells were infected with the mixture for 1 h, and then the supernatants were replaced with DMEM for 20 h. Total DNA was extracted from the PRV-infected cells, and RT-PCR was used to determine the PRV copy number.

Pre-treatment assay: PK15 cells were incubated in DMEM with or without T2 for 2 h, and then the supernatants were removed and replaced with DMEM containing PRV for 2 h. Next, the supernatants were replaced with DMEM for 20 h. Total DNA was extracted from the PRV-infected cells, and RT-PCR was used to determine the copy number of the PRV.

Virus entry assay: PK15 cells were incubated with PRV at 4 °C for 2 h. After removing the supernatants, DMEM with or without T2 was added to the plates, which were incubated at 37 °C for 2 h, allowing virus entry. Then, the supernatants were removed and DMEM was added to each plate for 20 h. Total DNA was extracted from the PRV-infected cells, and RT-PCR was used to determine the copy number of the PRV.

Virus replication assay: PK15 cells were infected with PRV for 2 h, and then the supernatants were replaced with DMEM with or without T2 for 20 h. Total DNA was extracted from the PRV-infected cells, and RT-PCR was used to determine the copy number of the PRV.

### 5.8. DNA and RNA Extraction and qPCR

The cells were divided into four groups (NC, T2, PRV, T2+PRV), as described in Section 2.3. After 20 h, we extracted total RNA from these groups using Trizol. About 1 μg of RNA specimen was used to generate cDNA using a Revert Aid Strand cDNA Synthesis Kit (Thermo Scientific, Waltham, MA, USA). The primer sequences used for qPCR are shown in Table 1. The qPCR procedure was conducted with the following steps: 95 °C for 5 min, followed by 39 cycles of 95 °C for 30 s, and then 60 °C for 30 s. The relative mRNA expression of oxidative-stress- and apoptosis-related genes was determined via qPCR using the β-actin gene as a loading control.

### 5.9. Western Blot

PK15 cells of the abovementioned groups were lysed with RIPA lysis buffer for 20 min at 4 °C. The supernatants were collected in sterile 1.5 mL microcentrifuge tubes after centrifugation at 14,000× *g* for 20 min at 4 °C. Equal amounts of protein determined by the BCA protein assay kit(Abcam, Shanghai, China) were electrophoresed and transferred onto nitrocellulose membranes. After blocking with 3% BSA for 1 h, the membranes were incubated with primary antibodies for 4 h, followed by incubation with secondary antibodies for 1 h. The membranes were washed five times with PBS after each antibody incubation. Finally, band signals on the membranes were visualized and analyzed using ChemiDoc XPS.

### 5.10. Statistical Analyses

Statistical analyses were performed using Graph Pad Prism 7.0 by one-way analysis of variance (ANOVA), and the data were expressed as the means ± SD. *p* < 0.05 was regarded as significant.

## Figures and Tables

**Figure 1 toxins-14-00121-f001:**
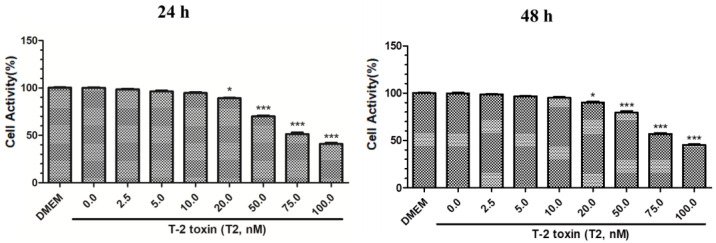
Effect of different T2 concentrations at 24 and 48 h on PK15 cell activity. * Significant difference compared to the control group (*p* < 0.05). *** Significant difference compared to the control group (*p* < 0.001).

**Figure 2 toxins-14-00121-f002:**
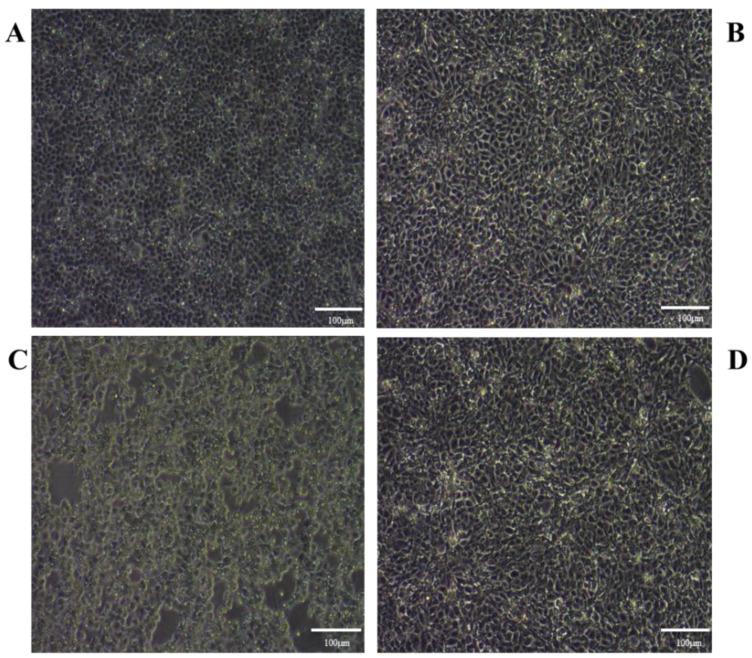
Cell morphology of PK15 cells in the different groups at 24 h. (**A**) PK15 cells. (**B**) PK15 cells treated with 10 nM T2. (**C**) PRV-infected PK15 cells (MOI = 0.01). (**D**) PRV-infected PK15 (MOI = 0.01) cells treated with 10 nM T2.

**Figure 3 toxins-14-00121-f003:**
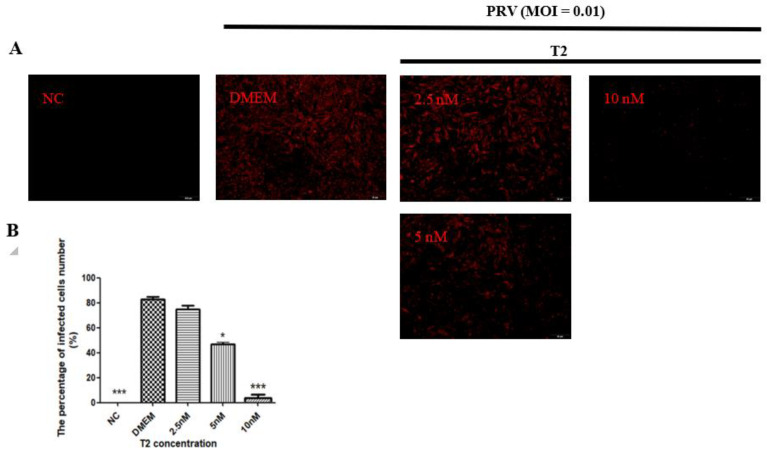
The number of PRV-infected (MOI = 0.01) cells in the different groups after 20 h. (**A**) The number of infected cells was detected by immunofluorescence assay. Red fluorescence indicated cells that were infected with PRV. (**B**) Quantification of the results in (A). * Significant difference compared to the DMEM group (*p* < 0.05). *** Significant difference compared to the DMEM group (*p* < 0.001).

**Figure 4 toxins-14-00121-f004:**
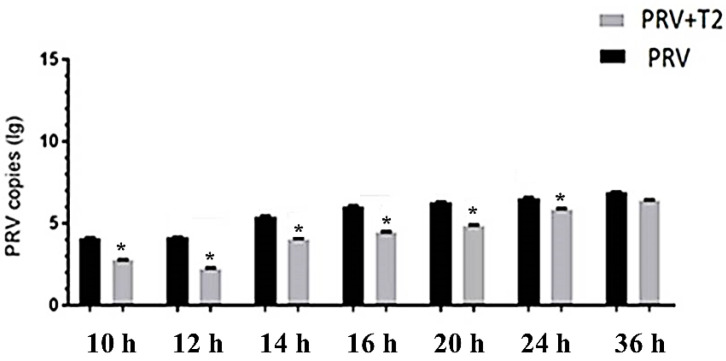
PRV copy number in PK15 cells in the different groups and at different time points (MOI = 0.01). * Significant difference compared to the PRV group (*p* < 0.05).

**Figure 5 toxins-14-00121-f005:**
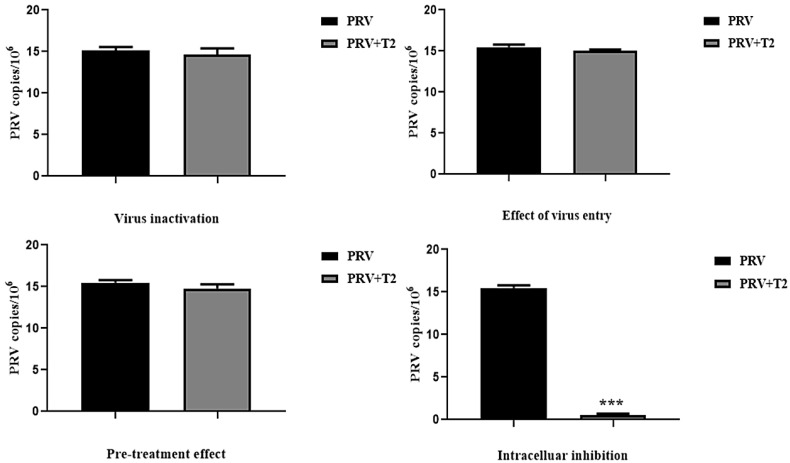
Influence of different treatment conditions of T2 on PRV infection. PK15 cells infected with PRV (MOI = 0.01) were treated with T2 (10 nM) under different treatment conditions: virus inactivation, pre-treatment effect, inhibition of virus entry, and intracellular inhibition. *** Significant difference compared to the PRV group (*p* < 0.001).

**Figure 6 toxins-14-00121-f006:**
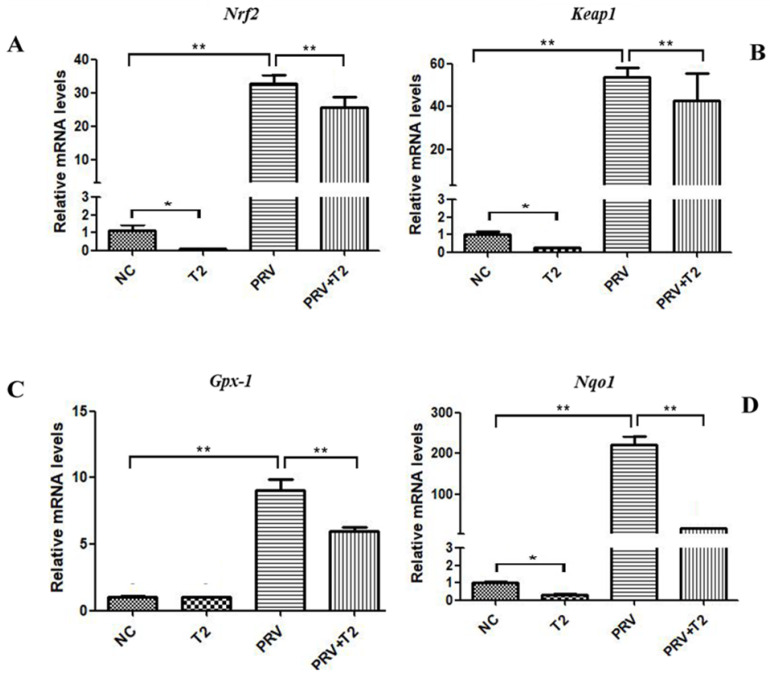
mRNA levels of oxidative-stress-related genes in the different groups after 20 h. (**A**) *Nrf2*. (**B**) *Keap1*. (**C**) *Gpx-1*. (**D**) *Nqo1*. * Significant difference (*p* < 0.05). ** Significant difference (*p* < 0.01).

**Figure 7 toxins-14-00121-f007:**
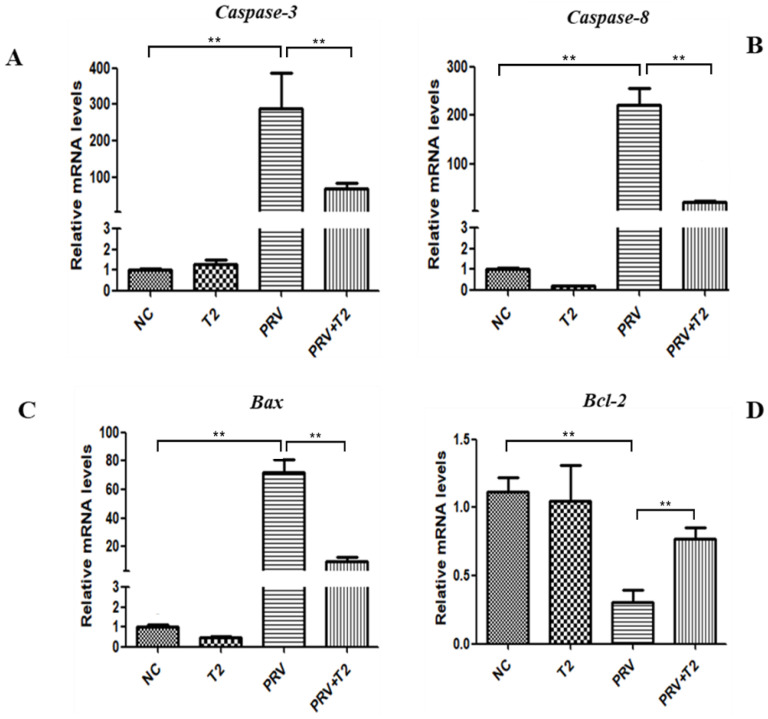
mRNA levels of apoptosis-related genes in the different groups after 20 h. (**A**) *Caspase-3*. (**B**) *Caspase-8*. (**C**) *Bax*. (**D**) *Bcl-2*. ** Significant difference (*p* < 0.01).

**Figure 8 toxins-14-00121-f008:**
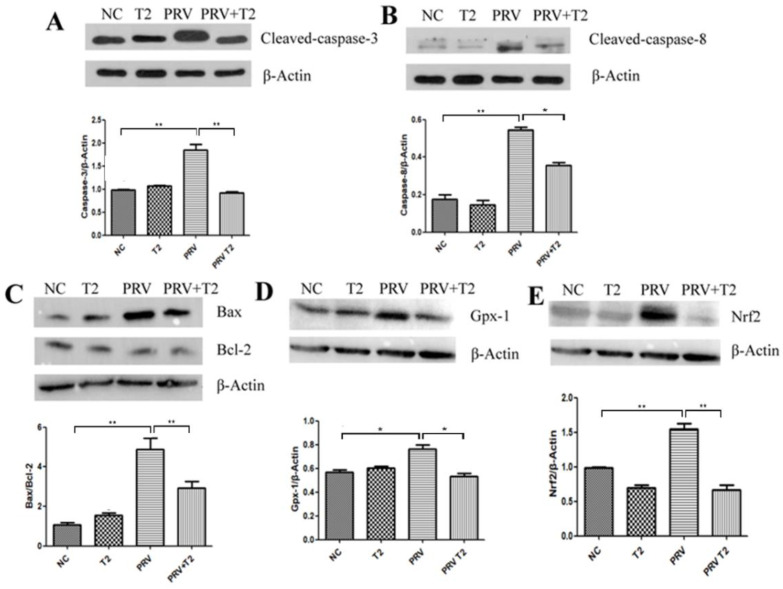
The expression of proteins related to oxidative stress and apoptosis signaling pathways in the different groups after 20 h. (**A**) Cleaved-caspase-3. (**B**) Cleaved-caspase-8. (**C**) Bax and Bcl-2. (**D**) Gpx-1. (**E**) Nrf2. Full-length blots are presented in Appendix A. * Significant difference (*p* < 0.05). ** Significant difference (*p* < 0.01).

**Table 1 toxins-14-00121-t001:** Information about primers.

Gene	Primer Sequence (5′-3′)
*PRV-gB*	F: AAGTTCAAGGCCCACATCTA
	R: TGAAGCGGTTCGTGATGG
*Caspase-3*	F: GGA ATG GCA TGT CGA TCT GGT
	R: ACT GTC CGT CTC AAT CCC AC
*Caspase-8*	F: TCT GCG GAC TGG ATG TGA TT
	R: TCT GAG GTT GCT GGT CAC AC
*Bax*	F: ATG ATC GCA GCC GTG GAC ACG
	R: ACG AAG ATG GTC ACC GTC GC
*Bcl-2*	F: GAA ACC CCT AGT GCC ATC AA
	R: GGG ACG TCA GGT CAC TGA AT
*Gpx-1*	F: CGTGCAACCAGTTTGGACAT
	R: AGCATGAAGTTGGGCTCGAA
*Nqo1*	F: GATCATACTGGCCCACTCCG
	R: GAGCAGTCTCGGCAGGATAC
*Nrf2*	F: CATAGCAGAGCCCAGTACCA
	R: AGGGGTTGGATTGGGTTTAGT
*Keap1*	F: ACTTTCGTAGCCCCCATGAA
	R: ATCCCTAGCGTGCAGGTGT
*β-actin*	F: CTG CGG CAT CCA CGA AAC T
	R: AGG GCC GTG ATC TCC TTC TG

## Data Availability

Not applicable.

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
