# Peer review of "Low-Concentration T-2 Toxin Attenuates Pseudorabies Virus Replication in Porcine Kidney 15 Cells"

_toxins, 2022, doi:10.3390/toxins14020121_

Round 1
Reviewer 1 Report
The article titled “Low-concentration T-2 toxin attenuates pseudorabies virus 2 replication in porcine kidney 15 cells”, is a study that provides a different perspective on the antiviral property of T-2 toxin. In this study, besides evaluating the cytotoxic effect of T-2 toxin on the cell, the effect of exposure of cells to this toxin during virus infection was evaluated by evaluating cellular oxidative stress and apoptosis biomarkers. The study design is consistent with the hypothesis. This study work acceptable with minor revision.
- Authors should give the IC50 value of T-2 toxin in the assessment of cytotoxicity
- It is known that t-2 toxin has serious toxic effects. In the discussion part of the study, if this toxin is a treatment candidate, its risk assessment should be given in detail.
- Figure 2,3 and 4 are not clear, the image quality of the pictures should be increased. Also, the bars are not visible in the pictures, the color can be changed.
Kind Regards

Author Response
Thank you very much for your suggestions on our article. Our reply is in the following word document. Please check it.

Reviewer 2 Report
As a reviewer, I cannot recommend this manuscript for publication. There are numerous problems in the manuscript as it is written.
The Introduction is sound, but must include the physiologic concentrations of T2 that pigs encounter in their feed. Are you cell culture experiments even within 2 orders of magnitude of what would be found in a pig exposed to T2 in its feed?
The Results section has problems with experimental design and presentation. Much more detail should be given for the experimental design in each figure legend or in the prose of the Results section. Reading this manuscript is a chore.
Figure 1: The plot is showing "Cell Activity" as a percentage. In the figure legend or in the Results text, you should describe the "Cell Activity" assay. Putting this in Methods alone is unacceptable.
The plot is a dose response plot. To pharmacologists and toxicologists, this would be most easily interpreted if presented in a X Y scatter plot instead of a bar graph. Also, different shades (or designs) of bars in a bar graph are usually reserved for denoting different treatments or background conditions. Use the same color for all bars if you decide to stick with the bar graph. I do not understand what the letters indicate in regards to significance. You need to define the letters.
Figure 2:
This is not a useful microscopy figure. PrV has an obvious nuclear CPE that could be shown with even the simplest microscopes at 20-40X. The MOI is not listed and MUST be noted for all PrV experiments. I would remove this figure. Since the microscopy is bad, it gives the reader no information that a viral titer experiment does not give. A plaque assay giving a supernatant titer with these timepoints and treatments would be more appropriate.
Figure 3:
There are two obvious typos in this figure (both the figure number and the panel letters). No MOI is indicated in the figure, legend, or results section. Again, a plaque assay with titers would be much more useful than this low-impact micrograph figure.
Figure 4:
You have not indicated the time of infection, MOI, or immunogen in the results or figure legend. It is not sufficient to have this information only in the methods. The MOI listed here is 0.1. This is counter to the 0.01 you have listed throughout the paper.
Figure 5:
I don't understand why you are showing copy numbers in two different graphs and you don't explain in the results. The MOI, etc should be listed here. The letters for significance don't make sense without a full key.
Figure 6:
Two sentences in the results is insufficient to describe four different assays.
Figure 7:
I'd recommend writing gene names above each graph.
Figure 8:
I'd recommend writing gene names above each graph.
For figures 6-9, you need to guide the reader through these genes. I'm not sure a two-fold (or less) reduction in transcripts of many of these oxidative stress/apoptosis genes should translate to the differences seen in PrV replication.
Other notes: you say that the PrV strains were acquired from dead piglets and that they compare to "traditional variant strains." What are these strains? Becker? Bartha? Were all the viral stocks in this study the same , or generated at different times? Were the piglets handled as part of this study? If so a statement of ethics must be included.
I'm also a little baffled by the lack of plaque assays, a staple of PrV virology. Are the differences in this paper solely discernable by qPCR?
Overall, the virological approach and presentation of this paper are lacking. This manuscript needs a lot of work to be considered for publication. You absolutely must have times, MOI, and etc listed for all figures. You should describe your experimental design for each figure.
Author Response
Thank you very much for your valuable comments and suggestions on our manuscript. Following the reviewers’ comments, we have modified and improved our manuscript according to your kind advices and referee’s detailed suggestions. Enclosed please find the responses to the referees. We sincerely hope this manuscript will be acceptable to be published on Toxins.The specific reply is in the following word document.

Round 2
Reviewer 2 Report
I thank the authors for revising the manuscript.
In my first review, I requested changes that I felt were reasonable.
- The physiologic concentration of T2 in pigs needed to be listed.
- The authors have listed this, and I thank them for that. They reveal that, by my back-of-the-envelope calculation, the concentration of T2 in an impacted pig would be about 1 nM. There is no critical analysis of this fact in the manuscript. The authors need higher doses (up to 10-fold more!) than what is in the feed to have an effect on PrV replication in PK15 cells. Research like this is valuable, but the dose-dependence should change the entire framework of the writing. You only get effects on PrV replication in experiments at a MUCH higher dose than is relevant in a veterinary setting.
- You are discussing a toxin in a journal called Toxins. You must get the relationship between dose and response through to readers. It is a requirement of the toxicology field.
- Generally, I asked for more details of how experiments were done in the results section or figure legends.
- I feel like this was largely ignored. I find most of the added sentences unhelpful. I apologize if there is a word limit (I don’t know this as a reviewer) but you can be much more scientifically specific in a sentence or two for EACH EXPERIMENT than you are being. I think you need to spend more effort on communicating your methods in the prose of the Results.
- Figure 1: I asked for a description of the “cell activity” assay. This should be very easy to describe in one sentence in the results or in the figure legend.
- The authors have added that it was an “MTT” assay. It still doesn't clear much up. I'm thinking something along the lines of "We performed an MTT assay by measuring the absorbance of MTT at XXX nm and compared the mictochondiral reductase activity in PK15 cells to a DMEM control." You can fit a lot of information in a single sentence that will help readers interpret results.
- In Figure 1, the “significance” letters were unclear
- The authors replaced the letters with asterisks in Figure 1, but left the undefined letters in all other figures! A reader won’t automatically know what a, b, c, or d means when describing significance! This should be uniform throughout the paper.
- I think the Figure 2 CPE images are not useful. It was not replaced. This is an editorial decision that has less impact on the science, so I defer to authors.
- Figure 4 (now 3): Correction to incorrectly listed MOI was made. The authors list the IF antigen, gB, in the results.
- In new Figure 5, I asked for more detail about what seems to be four different assays: Inactivation, pretreatment, entry, and intracellular inhibition. It’s unacceptable to not describe how each of these was done, briefly, in the results (as I have done above with MTT). I have no idea what you did in any case to perform the experiment and quantify copy numbers. It is not sufficient to only have this in the methods. Lines 165-167 are not sufficient.
- The authors have added more information about the genes assayed in Fig 6 and Fig7. I think that sentence on lines 177-178 is sufficient, but the new line 185 is not. The authors should use a few words to describe the caspases, Bax, and Bcl-2. They don’t exist to detect apoptosis; they have biological roles in apoptosis. As a group, those roles can likely be described in one sentence.
- I asked about the variant strain and source. The correction on lines 288-291 is sufficient. No ethics statement is needed for pigs since the stocks were in the freezer.
Overall, I’d like the authors to rethink this paper in a dose-response sense. Reframe your results and discussion to consider the problem from a veterinary perspective (even if the authors are not veterinarians). I’d also like much more effort in communicating the steps taken in the results. The Ms suffers from a lack of information and I feel like not much effort was made to improve this.
Editorially, I’d still suggest lots of work to change this paper. Especially if the lab is closed there may be some time to improve the writing and more effectively communicate what you did and what it means.
Good luck and take care.
Author Response
Thank you very much for your valuable comments and suggestions on our manuscript. Following the reviewers’ comments, we have modified and improved our manuscript according to your kind advices and referee’s detailed suggestions. Enclosed please find the responses to the referees. We sincerely hope this manuscript will be acceptable to be published on Toxins.
Details
Dear Editor and Reviewers,
Thank you very much for your valuable comments and suggestions on our manuscript. Following the reviewers’ comments, we have modified and improved our manuscript according to your kind advices and referee’s detailed suggestions. Enclosed please find the responses to the referees. We sincerely hope this manuscript will be acceptable to be published on Toxins.
Reply to Reviewer 2
We thank the reviewer for his/her constructive criticisms that have helped us to improve our manuscript. The point-by-point response to the comments is given below.
Thank you very much for all your help and looking forward to hearing from you soon.
Best regards
Sincerely yours
Prof. Yang
Please find the following Response to the comments of referees:
Question 1:
The physiologic concentration of T2 in pigs needed to be listed.
The authors have listed this, and I thank them for that. They reveal that, by my back-of-the-envelope calculation, the concentration of T2 in an impacted pig would be about 1 nM. There is no critical analysis of this fact in the manuscript. The authors need higher doses (up to 10-fold more!) than what is in the feed to have an effect on PrV replication in PK15 cells. Research like this is valuable, but the dose-dependence should change the entire framework of the writing. You only get effects on PrV replication in experiments at a MUCH higher dose than is relevant in a veterinary setting.
You are discussing a toxin in a journal called Toxins. You must get the relationship between dose and response through to readers. It is a requirement of the toxicology field.
Response to Question 1: I am very grateful to your comments for our manuscript. The reason for the selection of T2 concentration in vitro experiment is that: 1. 10nM T2 was the highest concentration screened by MTT assay which had no significant effect on the activities of PK15 cells; 2. According to published articles, the toxic doses of T2 were different between various pigs and porcine cells, and next we had analyzed and discussed the toxic dose of T2 in pigs in vivo and in vitro in accordance with your requirements [1-3]; 3. 10nM T2 could not decline the activities of PK15 cells in present study, and 10 nM T2 was rarely used in toxicity experiment in previous paper because the dosage of T2 mentioned above could not cause significant toxic effects; 4. Furthermore, there were no significant differences between control cells and the cells treated by only T2 in expression levels of mRNAs and proteins related to apoptosis and oxidative stress. Therefore, 10 nM T2 does not induce obvious apoptosis, oxidative stress and cell death.
Thanks for your suggestion about the subsequent animal experiments that will be designed according to the editor's point of view.
Question 2:
Generally, I asked for more details of how experiments were done in the results section or figure legends. I feel like this was largely ignored. I find most of the added sentences unhelpful. I apologize if there is a word limit (I don’t know this as a reviewer) but you can be much more scientifically specific in a sentence or two for EACH EXPERIMENT than you are being. I think you need to spend more effort on communicating your methods in the prose of the Results.
Response to Question 2: Thank you for your suggestions. According to your requirements, we had simply added the description of each experiment in the description of the results (line101-103 162-164 173-176). We have added necessary descriptions to the results according to your requirements. And we had descripted each experiment in detail in the Materials and Methods.
Question 3:
Figure 1: I asked for a description of the “cell activity” assay. This should be very easy to describe in one sentence in the results or in the figure legend.
The authors have added that it was an “MTT” assay. It still doesn't clear much up. I'm thinking something along the lines of "We performed an MTT assay by measuring the absorbance of MTT at XXX nm and compared the mictochondiral reductase activity in PK15 cells to a DMEM control." You can fit a lot of information in a single sentence that will help readers interpret results.
Response to Question 3: Thank you for your suggestions. I had added the detailed experimental description in the results according to your requirements (line 101-103).
Question 4: In Figure 1, the “significance” letters were unclear
The authors replaced the letters with asterisks in Figure 1, but left the undefined letters in all other figures! A reader won’t automatically know what a, b, c, or d means when describing significance! This should be uniform throughout the paper.
Response to Question 4: Thank you for your suggestions. We replaced the letter with asterisks to keep it be uniform throughout the paper (Figure 3 –Figure 8). A declaration was added to each legend as you requested. The significance is explained in detail in the results
Question 5: In new Figure 5, I asked for more detail about what seems to be four different assays: Inactivation, pretreatment, entry, and intracellular inhibition. It’s unacceptable to not describe how each of these was done, briefly, in the results (as I have done above with MTT). I have no idea what you did in any case to perform the experiment and quantify copy numbers. It is not sufficient to only have this in the methods. Lines 165-167 are not sufficient.
Response to Question 5: Thank you for your suggestions. We added a simple description in the result and rewritten the legend (line 173-176 184-188). And the mode of action assay we did was supported by literature[4].
Question 6: The authors have added more information about the genes assayed in Fig 6 and Fig7. I think that sentence on lines 177-178 is sufficient, but the new line 185 is not. The authors should use a few words to describe the caspases, Bax, and Bcl-2. They don’t exist to detect apoptosis; they have biological roles in apoptosis. As a group, those roles can likely be described in one sentence. And the mode of action assay we do is supported by literature
Response to Question 6: Thank you for your suggestions. We have made a brief introduction to caspases, Bax, and Bcl-2 according to your request (line 201-203).
References:
- Rafai, P.; Papp, Z.; Jakab, L. Biotransformation of trichothecenes alleviates the negative effects of T-2 toxin in pigs. Acta veterinaria Hungarica 2013, 61, 333-343, doi:10.1556/AVet.2013.025.
- Li, X.; Wang, X.; Liu, S.; Wang, J.; Liu, X.; Zhu, Y.; Zhang, L.; Li, R. Betulinic acid attenuates T-2 toxin-induced cytotoxicity in porcine kidney cells by blocking oxidative stress and endoplasmic reticulum stress. Comparative biochemistry and physiology. Toxicology & pharmacology : CBP 2021, 249, 109124, doi:10.1016/j.cbpc.2021.109124.
- Seeboth, J.; Solinhac, R.; Oswald, I.; Guzylack-Piriou, L. The fungal T-2 toxin alters the activation of primary macrophages induced by TLR-agonists resulting in a decrease of the inflammatory response in the pig. Vet. Res. 2012, 43, 35, doi:10.1186/1297-9716-43-35.
- Zhao, X.; Cui, Q.; Fu, Q.; Song, X.; Jia, R.; Yang, Y.; Zou, Y.; Li, L.; He, C.; Liang, X.; et al. Antiviral properties of resveratrol against pseudorabies virus are associated with the inhibition of IκB kinase activation. Scientific reports 2017, 7, 8782, doi:10.1038/s41598-017-09365-0.
